# High mortality in an outbreak of multidrug resistant *Acinetobacter baumannii* infection introduced to an oncological hospital by a patient transferred from a general hospital

Patricia Cornejo-Juárez[1], Miguel Angel Cevallos[2‡], Semiramis Castro-Jaimes[2‡], Santiago Castillo-Ramírez[2‡], Consuelo Velázquez-Acosta[3], David Martínez-Oliva[1], Angeles Pérez-Oseguera[2‡], Frida Rivera-Buendía[1], Patricia Volkow-Fernández[1]*

**1** Departamento de Infectología, Instituto Nacional de Cancerología, Ciudad de México, México, **2** Programa de Genómica Evolutiva, Centro de Ciencias Genómicas, Universidad Nacional Autónoma de México, Cuernavaca, Morelos, México, **3** Laboratorio de Microbiología, Instituto Nacional de Cancerología, Ciudad de México, México

☯ These authors contributed equally to this work.
‡ These authors also contributed equally to this work.
* pvolkowf@gmail.com

**Data Availability Statement:** Data are available from the Comité de Etica en Investigación and

## Abstract

### Objective

To describe the clinical features, outcomes, and molecular epidemiology of an outbreak of multidrug resistant (MDR) *A. baumannii*.

### Methods

We performed a retrospective analysis of all MDR *A. baumannii* isolates recovered during an outbreak from 2011 to 2015 in a tertiary care cancer hospital. Cases were classified as colonized or infected. We determined sequence types following the Bartual scheme and plasmid profiles.

### Results

There were 106 strains of *A. baumannii* isolated during the study period. Sixty-six (62.3%) were considered as infection and 40 (37.7%) as colonization. The index case, identified by molecular epidemiology, was a patient with a drain transferred from a hospital outside Mexico City. Ninety-eight additional cases had the same MultiLocus Sequence Typing (MLST) 758, of which 94 also had the same plasmid profile, two had an extra plasmid, and two had a different plasmid. The remaining seven isolates belonged to different MLSTs. Fifty-three patients (50%) died within 30 days of *A. baumannii* isolation: 28 (20%) in colonized and 45 (68.2%) in those classified as infection (p<0.001). In multivariate regression analysis, clinical infection and patients with hematologic neoplasm, predicted 30-day mortality. The molecular epidemiology of this outbreak showed the threat posed by the introduction of MDR strains from other institutions in a hospital of immunosuppressed patients and

Comité de Investigación (contact by phone +525556280400 ext. 37015 or email: comite. cientifico.incan@gmail.com) for researchers who meet the criteria for access to confidential data. An anonymized data set are available within the paper and its Supporting Information files.

**Funding:** MAC, SCR:This project was partially financed by PAPIIT200318 (PAPIIT:[Programas de Apoyo a Proyectos de Investigación e Innovación Tecnológica]-Support program for research and technological innovation projects). Universidad Nacional Autónoma de México.

**Competing interests:** The authors have declared that no competing interests exist.

highlights the importance of adhering to preventive measures, including contact isolation, when admitting patients with draining wounds who have been hospitalized in other institutions.

## Introduction

*Acinetobacter baumannii* has become a major hospital pathogen, due to multidrug resistant (MDR) strains and it is now considered one of the six most important microorganisms that causes hospital-acquired infections worldwide, with attributable mortality ranging from 8% to 35% according to strain and type of infection as well as increase in hospital stays and health care expenditures [1,2]. Bloodstream infection (BSI) and pneumonia, especially ventilator-associated pneumonia (VAP), are the most severe infections caused by this gram-negative bacterium [3,4].

Carbapenems have been considered the drugs of choice to treat *A. baumannii* infections. These drugs are still the first-line agents for empirical therapy in areas with high rates of susceptibility [3]. However, the extensive use of carbapenems for the treatment of cephalosporin-resistant *Enterobacteriaceae* has been regarded as one of the main risk factors promoting the emergence and spread of multidrug-resistant (MDR) *A. baumannii* strains.

The role of the environmental contamination in the transmission of nosocomial infections in general and in *A. baumannii* infections in particular is well recognized [5]. This pathogen is able to grow at various temperatures and pH conditions, so it has the ability to persist in either moist or dry conditions in the hospital environment, thereby contributing to transmission [6, 7, 8]. We documented an outbreak of an MDR *A. baumannii* strain identified initially in 2011 in the intensive care unit (ICU) that lasted until 2015. The strain was introduced by an infected patient transferred from another hospital. This study aimed to identify the epidemiology of the outbreak and to describe the clinical evolution and risk factors for adverse outcomes of patients during the outbreak of MDR *A. baumannii*.

## Methods

We conducted a retrospective study at Instituto Nacional de Cancerología (INCan), a 135-bed tertiary care oncology hospital located in Mexico City, with an average of 170,000 medical visits and 7,500 hospital discharges per year, taking care mainly of uninsured patients, and patients covered by the "popular insurance" design to offer a basic package of services for certain malignancies. The ICU has six beds, with a mean of 230 hospitalizations per year; 58% of them are surgical patients with solid tumors, 25% have hematologic malignancies, and 17% are patients with solid tumors on chemotherapy.

In March 2011, we isolated the first strain of MDR *A. baumannii*; since then, multiple isolates from different sources of hospitalized patients at the ICU were recovered. No MDR *A. baumanii* strains had been isolated previously in the hospital. During 2011 to 2012, 78 isolates were identified (73.6%), decreasing the number of isolates recovered in the coming three years, until 2015. From January 2011 to December 2015, 106 patients with MDR *A. baumannii* isolates were identified, all were included in the study. Each isolate was classified as infected or colonized according to the criteria of two independent infectious diseases (ID) clinicians. A case was defined when the patient exhibited signs and symptoms of infection with or without a systemic inflammatory response. Bloodstream infection (BSI) was defined as laboratory-confirmed isolation of *A. baumannii* from blood cultures. Pneumonia was considered when new

or progressive infiltrates on x-ray appeared with at least two of the following parameters: fever (>38˚C) or hypothermia (<35.5˚C), leukocytosis (>12,000 cells/ml) or leukopenia (<4000 cells/ml), and positive bronchial aspirate culture [9]. Urinary tract infection was considered in those patients who present dysuria or hematuria and frequency or urgency, and urinalysis with pyuria and bacteriuria or positive nitrites. Surgical site infection defined as the infection in the part of the body where the surgery took place and occurring within 30 days after the operation. Airway colonization was defined if *A. baumannii* was isolated without symptoms of infection, and did not receive specific antimicrobial therapy. If there was a disagreement between the two physicians, a third ID physician analyzed the case, and a consensus was reached.

Clinical information was collected from the medical charts, including type of oncological disease, clinical stage (recent diagnosis, progression, relapse, partial or complete remission), recent chemotherapy or surgery (less than one month), recent hospitalization or antimicrobial use (previous 3 months), length of hospital stay, ICU admission and length of ICU stay, SOFA (Sequential Organ Failure Assessment) at ICU admission and at the date from *A. baumannii* culture was taken, days on mechanical ventilation, antimicrobial regimen, and outcome at 72 hours and at 30 days (alive, death attributable to infection, death related to neoplasm, or death due to other cause). Appropriate antimicrobial regimen was considered if it included colistin or tigecycline for ≥48 hours within the first 24 hours from isolation.

*A.baumannii* was isolated from clinical specimens submitted to the microbiology laboratory; identification and antimicrobial susceptibility were determined using the Gram-Negative Identification Panel (BDPhoenix^TM automated system, New Jersey, USA). Identification of genotypic strains was confirmed by matrix-assisted laser desorption and ionization-time of flight mass spectrometry (MALDI-TOF-MS; Microflex, USA). MDR strains were tested for antibiotic disks on Mueller-Hinton agar containing colony suspension of *A. baumannii* equivalent to a 0.5 McFarland, Clinical and Laboratory Standards Institute (CLSI) recommendations for susceptibility [10]. Antibiotic disks included ceftriaxone (30 μg), ceftazidime (30 μg), ciprofloxacin (5 μg), imipenem (10 μg), meropenem (10 μg), gentamicin (10 μg), and sulfamethoxazole/trimethoprim (SXT-1.25/23.75 μg) (CLSI). Colistin (10 μg) disks were tested although no parameters had been determined by CLSI. All isolates had an inhibition zone diameter ≥21 mm. [10]. Strains are considered MDR if they showed resistance to more than one agent in ≥3 antimicrobial categories such as aminoglycosides, antipseudomonal carbapenems and/or cephalosporins, fluoroquinolones, beta-lactam-beta-lactamase inhibitors, folate pathway inhibitors, and tetracyclines (Magiorakos criteria) [11]. All the strains included in this study, were resistant to all previous antibiotics families except tetracyclines and colistin. MICs reported were: ceftriaxone >32, ceftazidime >2, ciprofloxacin >2, imipenem >8, meropenem >8, gentamicin >8, and SXT >2/38. Strains were preserved and stored in glycerol at -70˚C for further analysis.

DNA was extracted from a representative strain of each pattern (Genomic DNA Purification Kit, Thermo Scientific) for subsequent PCRs. First, rpoB's Zone-1 [12] was amplified and sequenced to confirm taxonomic status. Plasmid profiles of each isolate were obtained during the study period with the modified Eckhardt protocol (S1 File) [13,14]. Each plasmid pattern was given an arbitrary roman numeral and used to group the strains. Strains confirmed as *A. baumannii* by their best hit against the NCBI database (at least 99% identity) were subject to MultiLocus Sequence Typing (MLST) scheme (Bartual) [15]. All PCR fragments were cloned in pJET1.2 (Thermo Scientific) and subject to Sanger sequencing with pJET universal oligonucleotides. *A. baumannii* MLST sequences were queried against the pubMLST database (http://pubmlst.org) to get their sequence type (ST). An ST number was requested to pubMLST for each one of the isolates with a new MLST profile.

## Statistical analysis

Categorical variables were described by frequencies and percentages. Quantitative variables were described by means, standard deviation, medians, and quartiles according to distribution. Univariate analysis was performed using Student's *t*-test or Mann-Whitney U test for continuous variables, and Chi-square or Fisher's exact test was used to compare categorical variables. Variables with P values ≤0.5 in the univariate analysis were included in the multivariate analysis. Rates of overall survival were estimated using the Kaplan Meier method and log rank test. P values <0.05 were considered statistically significant. Data were analyzed using STATA (ver. 14) software.

## Ethics

The study was approved by the INCan Ethics Review Board (Rev/0009/19). No consent form was required; patient´s information was anonymous and de-identified prior to analysis.

## Results

During the study period, 108 patients with MDR *A. baumannii* isolates were identified: two were excluded because *rpoB* sequencing one was identified as *A. haemolyticus* and one as *Acinetobacter spp*. There were included 106 patients with isolates from the following sources: blood (n = 18), bronchial aspirates (n = 49), surgical site infection (n = 20), urine (n = 12), pleural fluid (n = 4), biopsies (n = 2) and catheter tip (n = 1). Fourteen patients had MDR-*A. baumannii* isolated from two different clinical sites and three patients had the isolation from three different sites. Forty-one strains were isolated in 2011 (37.7%), 37 in 2012 (34.9%), 12 in 2013 (11.3%), 13 in 2014 (12.2%), and 3 in 2015 (2.83%). Sixty-six (62.3%) isolates were considered as infection and 40 (37.7%) were classified as colonization (Fig 1).

The index case was identified by molecular epidemiological analysis; was an infected hemato-oncological patient with a pleural drainage, who was transferred from another hospital outside Mexico City.

Of all the isolates obtained during the outbreak, 99 had the same sequence type MLST 758; 94 had the same plasmid profile as the one recovered from the index case, two had the same plasmid an extra plasmid, and two others were also MLST 758 but had different plasmid profiles. Seven had different MLST (four of them proceeded from infected patients and had been hospitalized in another hospital within the previous 60 days).

Representative plasma profiles of selected *Acinetobacter* strains are shown in Fig 2.

From the whole group, there were 59 men (55.6%); the mean age was 48.3 ± 16 years. Half of patients had a hematologic malignancy (*n* = 52, 49.1%), 66 (62.3%) had recent cancer diagnosis, 44 (41.5%) were receiving chemotherapy, and 28 (26.4%) had major surgery. Hematologic malignancies were more common in the infected group (76.9%) than in the colonization group (30%, *p* = 0.002). Also, there were more patients with hematologic malignancies classified as infected (60.6%) compared with patients with solid tumors (39.4%, *p* = 0.002). In the infected group, there were more patients (28.8%) who had received broad-spectrum antimicrobials during the previous three months vs. the colonization group (17.5%, *p* = 0.048). Other clinical and demographic characteristics are shown in Table 1.

Sixty-six patients (62.2%) were hospitalized in the ICU; the most common causes of ICU admission were: septic shock in 25 patients (37.9%) (eight related to *A. baumannii* infection, median of 12 days of hospitalization), and respiratory failure in 20 patients (30.3%). SOFA score at admission was not different between infected vs. colonized patients (*p* = 0.344); however, it was significantly higher in infected vs. colonized patients at the day in which the sample with *A. baumannii* isolation was taken (median 7 vs. 0, p < 0.001). Sixty-six patients (62.3%)

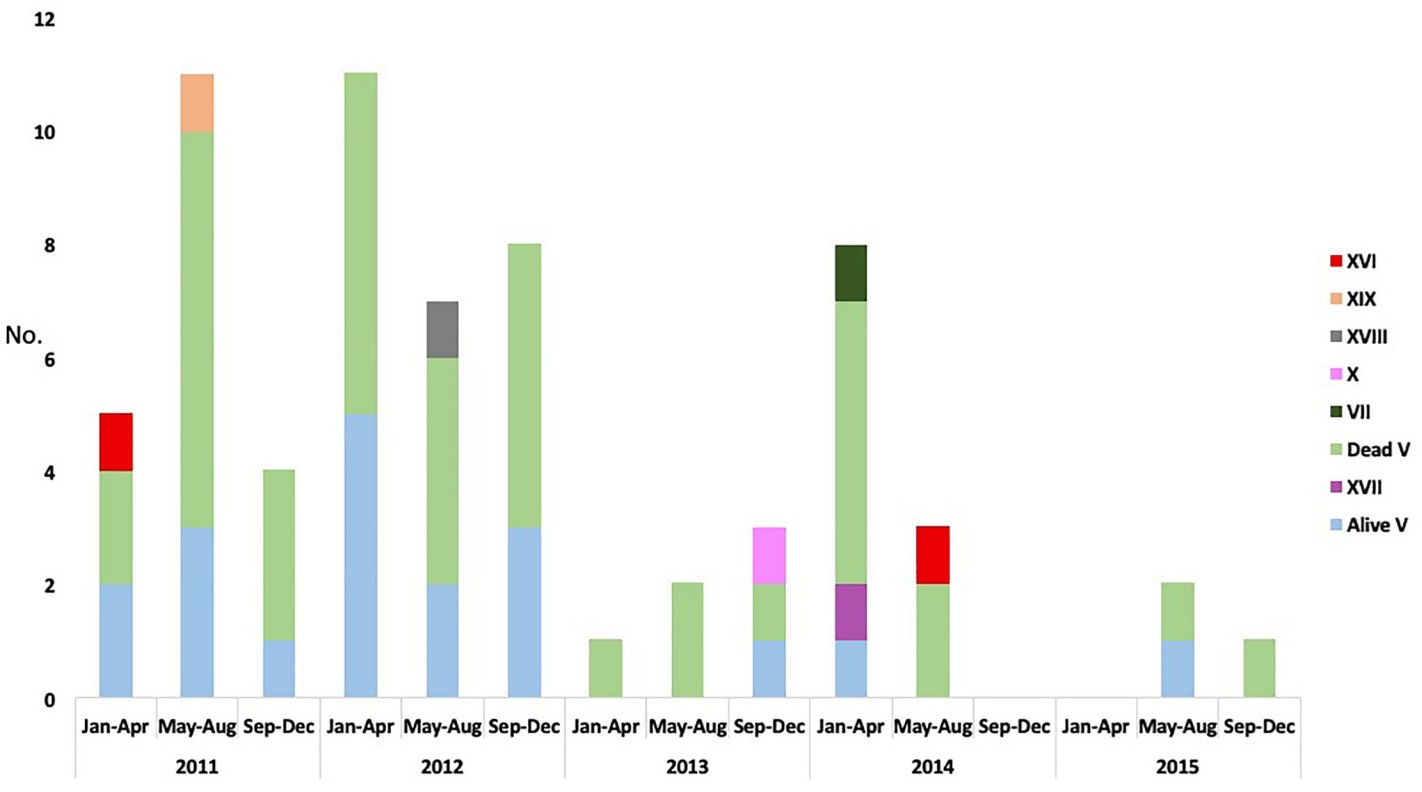

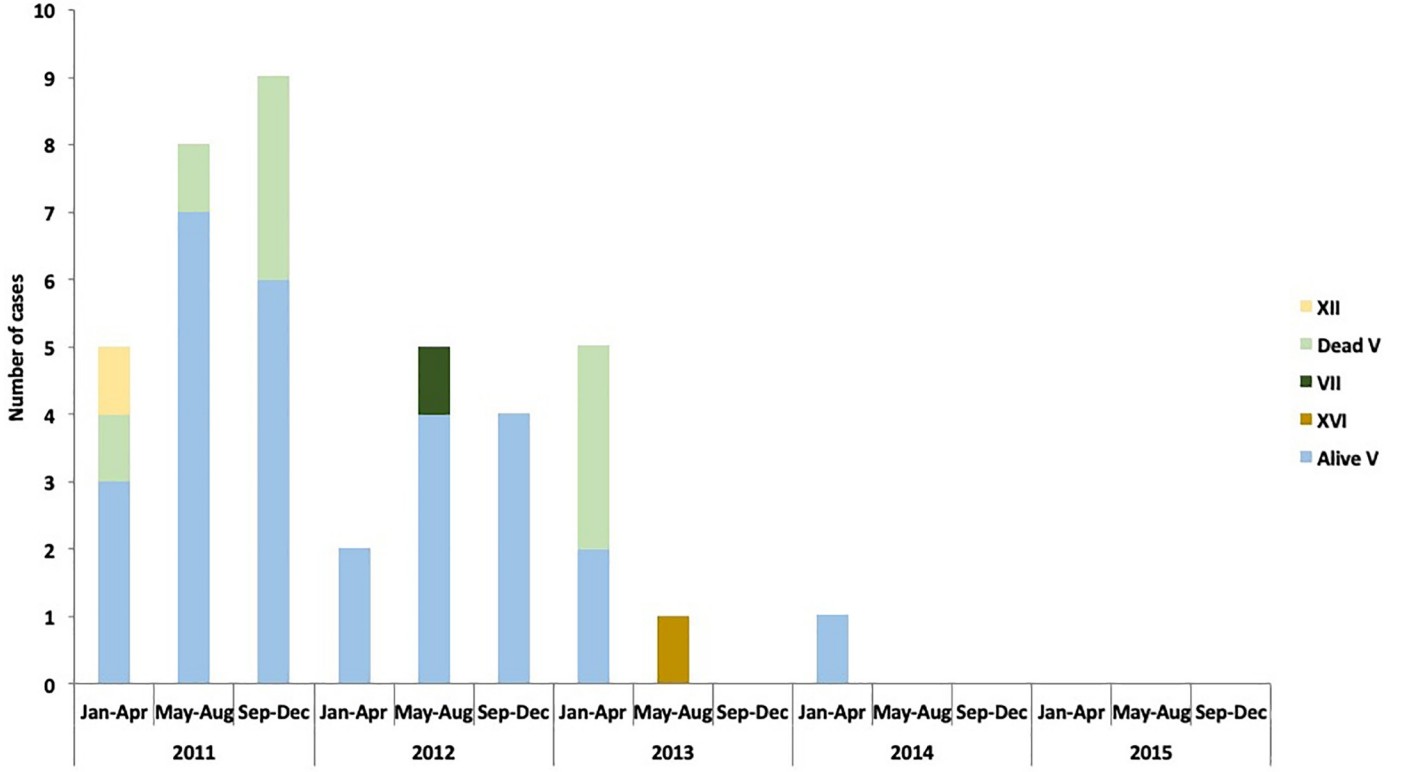

**Fig 1.** Number of infected (1a) and colonized (1b) patients during the study period (2011–2015) divided into alive or dead, including the plasmid profile.

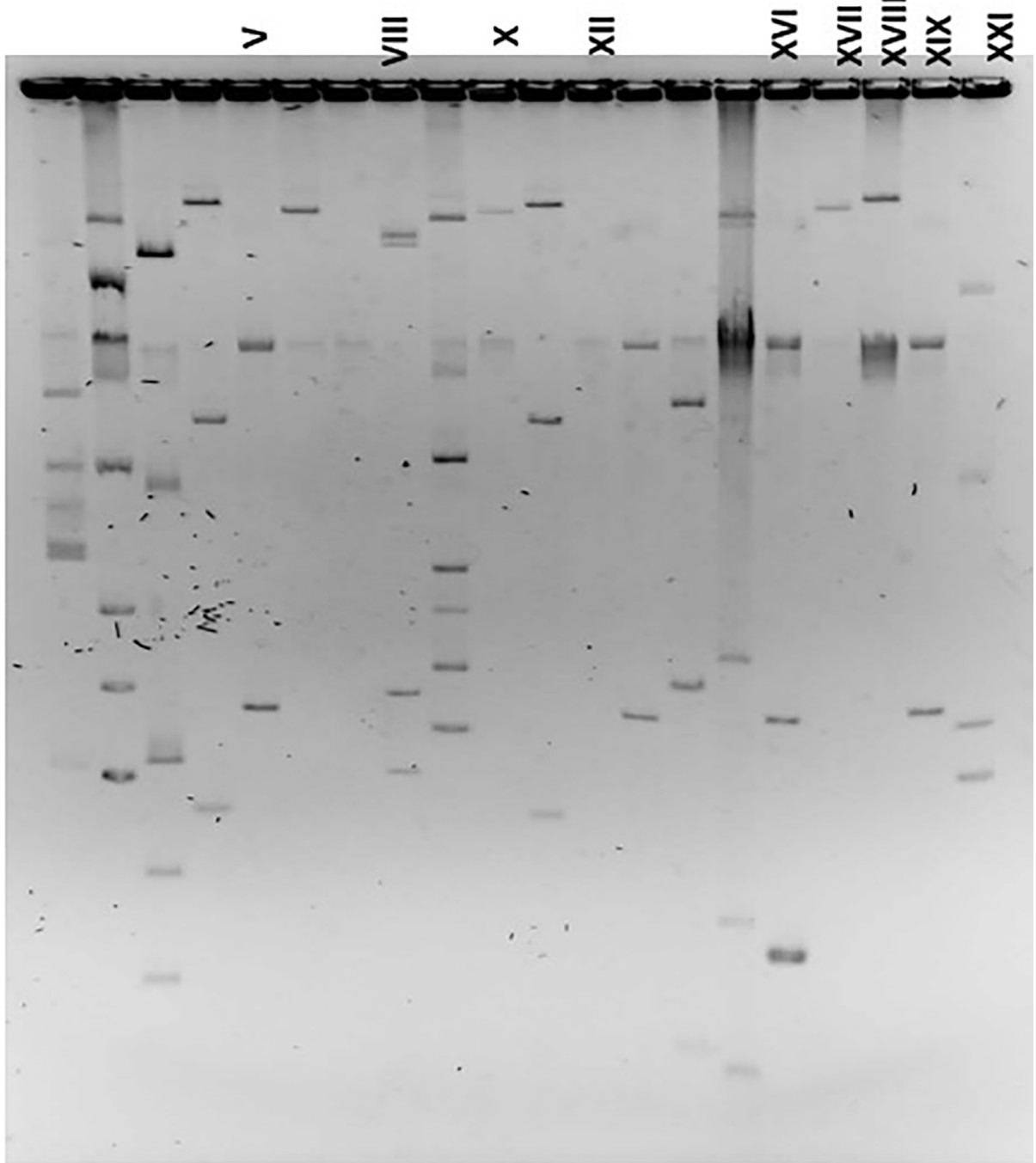

**Fig 2. Plasmid profiles of *Acinetobacter* strains.** Lanes marked with roman numerals are plasmid profiles isolated from *Acinetobacter baumannii*. Unmarked lanes are plasmid profiles of other *Acinetobacter* species isolated during this study period. Plasmid profile (strain number): V (2161), VIII (2339), X (3108), XII (3496), XIII (5189), XVI (4197), XVII (6793), XVIII (2275), XIX (2663), XXI (3275).

were on mechanical ventilation, 37 (56.1%) developed VAP, with a median of 8 days (IQR 3, 20 days) from intubation until *A. baumannii* isolation. Table 2.

Twenty-one patients (19.8%) died within 72 hours after *A. baumannii* isolation; 20 (95.2%) were classified as infected. During the first month, 53 patients (50%) died: 8 (20%) in the

**Table 1. Demographic and general characteristics of 106 patients with MDR-** *A. baumannii*, **classified as colonized or infected.**

| Characteristics N (%) | Total–N = 106 | Colonization–n = 40 (37.7) | Infection–n = 66 (62.3) | P |
|---|---|---|---|---|
| Age (years)[a] | 48.3 ± 16 | 51.1 ± 16 | 46.5 ± 15.9 | 0.156 |
| Male gender | 59 (55.7) | 21 (52.5) | 38 (57.6) | 0.610 |
| Solid tumor | 54 (50.9) | 28 (70) | 26 (39.4) | 0.002 |
| Hematologic malignancy | 52 (49.1) | 12 (30) | 40 (60.6) | |
| Oncologic status | | | | 0.393 |
| Recent diagnosis | 66 (62.3) | 23 (57.5) | 43 (65.2) | |
| Progression | 13 (12.2) | 5 (12.5) | 8 (12.1) | |
| Relapse | 12 (11.3) | 4 (10) | 8 (12.1) | |
| Complete remission | 9 (8.5) | 6 (15) | 3 (4.5) | |
| Partial remission | 6 (5.7) | 2 (5) | 4 (6.1) | |
| Recent chemotherapy[b] | 44 (41.6) | 14 (35) | 30 (45.5) | 0.289 |
| Recent radiotherapy[c] | 10 (9.4) | 2 (5) | 8 (12.1) | 0.311 |
| Previous hospitalization[b] | 55 (51.9) | 20 (50) | 35 (53) | 0.762 |
| Median length of previous hospitalization[d] (days) | 10 (5,17) | 9 (4, 218) | 10 (5,15) | 0.881 |
| Recent use of antimicrobials[b] | 26 (24.5) | 7 (17.5) | 19 (28.8) | 0.048 |
| Severe neutropenia | 16 (15.1) | 5 (12.5) | 11 (16.7) | 0.780 |
| Major surgery | 28 (26.4) | 16 (40) | 12 (18.2) | 0.013 |

[a]Mean ± standard deviation.

[b]During the previous three months.

[c]During the previous 6 months.

[d]Median (IQR).

colonized group and 45 (68.2%) in the infected group ($p$ <0.001). (Fig 3). From the infected patients who died, 16/18 (88.9%) had BSI, 22/30 (73.3%) had pneumonia, and 7/18 (38.9%) had *A. baumannii* isolated from another site ($p$ = 0.003). The time between *A. baumannii* isolation and death was shorter among patients whose strain was isolated from blood (median 3 days; IQR 2,5), compared with patients whose strain was isolated only from bronchial aspirates (median 8 days; IQR 3, 22), ($p$ = 0.03).

Considering only the infected patients, 32 (48.5%) received an appropriate antibiotic against *A. baumannii*, of which 29 received colistin (four as monotherapy, fifteen combined with rifampicin, two with meropenem, eight with meropenem and rifampicin), the mean days of colistin were 12 ± 6 days (no differences between patients with BSI, with VAP or with other infection). Two patients received tigecycline (one as monotherapy and one combined with meropenem), for 7 and 14 days each one. Eighteen from 32 patients who received appropriate treatment died in the first 30 days (56.2%), compared with 27 from 34 patients (79.4%) who did not receive an appropriate treatment ($p$ = 0.06).

In univariate analysis, infection vs. colonization, hematologic malignancies and inappropriate antimicrobial treatment were the risk factors significantly associated with 30-day mortality. In the multivariate regression analysis, infected patients and inappropriate antimicrobial treatment were the risks factors associated with death. Table 3.

## Discussion

We describe an outbreak that occurred from 2011 to 2015 (92.4% of the strains with MLST 758), peaking during 2011 and 2012 in the ICU, but persisting during the following years (2013 to 2015), including strains from patients hospitalized in wards outside the ICU. The

**Table 2. Clinical characteristics related to *A. baumannii* isolation in 106 patients, classified as colonized or infected.**

| Characteristics | Total | Colonization | Infection | P |
|---|---|---|---|---|
| | n = 106 (%) | n = 40 (37.7) | n = 66 (62.3) | |
| Site of *Acinetobacter* isolation | | | | <0.001 |
| Blood cultures | 18 (17) | 0 | 18 (27.3) | |
| Bronchial aspirate | 49 (46.2) | 19 (47.5) | 30 (45.4) | |
| Surgical site infection | 20 (18.9) | 15 (37.5) | 5 (7.6) | |
| Urine | 12 (11.3) | 3 (7.5) | 9 (13.6) | |
| Other | 7 (6.6) | 3 (7.5) | 4 (6.1) | |
| Hospital length before *A. baumannii* isolation (days)[a] | 17.6 ± 13.6 | 21.6 ± 17.1 | 16.3 ± 12.51 | 0.136 |
| ICU hospitalization[b] | 66 (62.3) | 20 (50) | 46 (69.7) | 0.046 |
| ICU stay before *A. baumannii* isolation (days)[a] | 10.1 ± 8.9 | 12 ± 11.4 | 9.3 ± 7.7 | 0.297 |
| ICU whole length stay (days)[a] | 15.9 ± 12.6 | 15.8 ± 12.5 | 16 ± 12.8 | 0.931 |
| SOFA at ICU admission[acd] | 8 (5, 11) | 7 (5, 10) | 8 (6, 11) | 0.344 |
| SOFA at *A. baumannii* isolation | 3 (0, 9) | 0 (0, 2) | 7 (3,11) | <0.001 |
| Mechanical ventilation (MV) | 66 (62.3) | 19 (47.5) | 47 (71.2) | 0.01 |
| Mean length of MV (days)[a] | 15.2 ± 12.5 | 15.8 ± 12.4 | 15 ± 12.7 | 0.828 |
| *A. baumannii* isolated from bronchial aspirate after 48 h of MV[e] | 37 (56.1) | 14 (73.7) | 23 (48.9) | 0.100 |
| Days of MV until isolation[a] | 12.6 ± 9.2 | 15.6 ± 12 | 10.8 ± 6.7 | 0.128 |
| Days from *A. baumannii* isolation until discharge[d] | 8 (3, 20) | 7 (4, 14) | 9 (3, 22) | 0.154 |
| Mortality at 72-hours | 21 (19.8) | 1 (2.5) | 20 (30.3) | <0.001 |
| Mortality at 30-day | 53 (50) | 8 (20) | 45 (68.2) | <0.001 |

[a]Mean ± standard deviation.

[b]ICU: Intensive Care Unit.

[c]SOFA (Sequential Organ Failure Assessment) at ICU admission were calculated in 66 patients: 20 colonized and 46 infected.

[d]Median (interquartile range).

[e]Percentage was obtained from the total of patients with mechanical ventilation.

strain was introduced into the institution by a patient (index case) with non-Hodgkin lymphoma transferred from another clinic outside Mexico City. MDR *A. baumannii* was isolated at arrival from the insertion site of a pleural tube. Initially, it was considered as a contaminant, and no treatment was prescribed. One week later, he developed respiratory failure, required mechanical ventilation, was admitted to the ICU and broad-spectrum antibiotics were administered. Three weeks after admission he received chemotherapy and seven days later he developed febrile neutropenia and MDR *A. baumannii* bacteremia was documented. He received colistin for 14 days and was transferred to another hospital still with clinical signs of serious infection. It has been demonstrated that *A. baumannii* is able to overcome specific and general host defense systems, and survive in contact with human host fluids and tissues [16].

In this series, we included 99 patients with an MLST 758 strain: 95 with the same plasmid profile (V), two with plasmid XVI- very similar to V, and two with another plasmid profile. The remaining seven isolates belonged to different MLSTs with different plasmids patterns; all of them have been hospitalized recently. They were probably colonized with the MDR strain when they arrived, and it became invasive during hospitalization due to therapeutic procedures.

This MLST has been identified previously in Mexican hospitals and in other countries of Latin America [17–19]. The MLST758 is an understudied lineage that does not belong to the international clones I and II; it has spread to Canada, Mexico, Honduras, and Colombia. This lineage represents a source of genetic diversity of MDR and XDR isolates that have not been studied [20, 21].

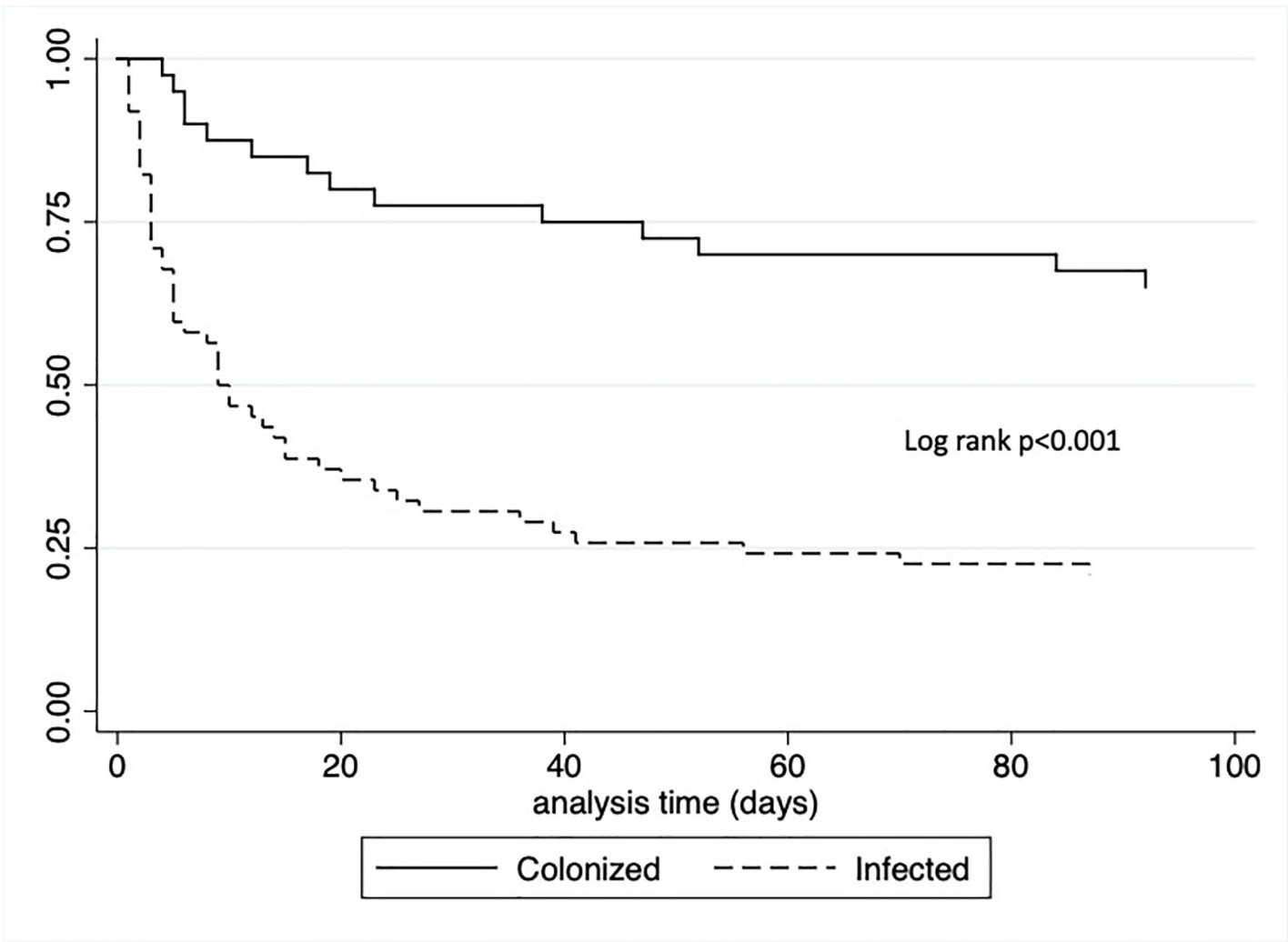

**Fig 3. Kaplan Meier survival in 106 patients with *A. baumannii* divided by colonized (n = 40) or infected (n = 66).**

Currently, colistin seems to be the most reliably effective drug in vitro against MDR *A. baumannii* [1, 3]. The efficacy of this antibiotic in severe infections caused by *A. baumannii* has been demonstrated in several retrospective and prospective series including patients with serious infections such as pneumonia and bacteraemia [3].

In this study, 32 of the 66 patients classified as infected who received appropriate antimicrobial treatment had lower mortality (56.2%), than those who did not receive appropriate antimicrobial therapy (79.4%), although it was not statistically significant. There were two reasons why patients did not receive appropriate treatment:1) 21 patients died within 48 hours after culture was taken, and MDR *A. baumannii* had not been reported yet; 2) the other thirteen died because colistin was not regularly available in our hospital during 2011 and the first half of 2012. This high mortality in the first hours is related to extremely rapid clinical progression described in cases with *Acinetobacter* infection [22].

The last strain (isolated in 2015) was from a patient that already was infected when he arrived from another hospital from the west coast of Mexico. The plasmid profile was similar to the outbreak strain; we presume it was reintroduced to our hospital but did not spread further.

**Table 3. Uni- and multivariate analysis for 30-day mortality in patients MDR- *A. baumannii*.**

| Characteristics—N (%) | Alive (n = 53, 50%) | Death (n = 53, 50) | Univariate | | Multivariate | |
|---|---|---|---|---|---|---|
| | | | OR (95% CI) | P | OR (95% CI) | P |
| Age ≤60 (years) | 44 (83) | 40 (75.5) | 1.6 (0.55–4.68) | 0.473 | - | |
| Age ≥60 | 9 (17) | 13 (24.5) | | | | |
| Colonized | 32 (60.4) | 8 (15.1) | 8.6 (3.12–24.86) | <0.001 | 1 | <0.001 |
| Infected | 21 (39.6) | 45 (84.9) | | | 6.58 (2.34–18.5) | |
| Solid tumor | 36 (67.9) | 18 (33.9) | 4.1 (1.7–10.1) | <0.001 | 1 | 0.03 |
| Hematologic malignancy | 17 (32.1) | 35 (66.1) | | | 2.12 (1.09–9.35) | |
| Recent diagnosis/CR[a] | 40 (75.5) | 36 (67.9) | 1.45 (0.57–3.73) | 0.388 | - | |
| Progression/relapse | 13 (24.5) | 17 (32.1) | | | | |
| No recent chemotherapy | 33 (62.3) | 29 (54.7) | 1.36 (0.58–3.19) | 0.43 | - | |
| Recent chemotherapy[b] | 20 (37.7) | 24 (45.3) | | | | |
| Appropriate treatment[c] | 8 (15.1) | 29 (54.7) | 2.9 (0.89–96) | 0.06 | 1 | 0.134 |
| Non-appropriate treatment | 13 (24.5) | 16 (30.2) | | | 2.03 (0.8–5.15) | |
| SOFA score <10[d] | 48 (90.6) | 37 (69.8) | 3.78 (1.16–14.37) | 0.02 | 1 | 0.525 |
| SOFA score >10 | 5 (9.4) | 16 (30.2) | | | 1.51 (0.42–5.41) | |

[a]Recent oncology diagnosis or complete remission.

[b]Chemotherapy in the previous month.

[c]Analysis was performed only in those patients considered as infected.

[d]SOFA score. Sequential Organ Failure Assessment calculated when MDR-*A. baumannii* culture was taken.

MDR *A. baumannii* isolation has increased during the last decades. In the 1970s, the majority of strains were sensitive to the commonly used antibiotics, but by 2007, up to 70% of isolates in certain settings were MDR, including some with resistance to carbapenems. Resistance is usually due to combined mechanisms, commonly including cell membrane impermeability, increased expression of efflux pumps, and production of beta-lactamases, commonly OXA type [23]. Exposure of *A. baumannii* to the selective pressure of potent antimicrobials has gradually led to a global prevalence of strains that are resistant to all beta-lactams, including carbapenems [23].

Recently, employing a population genomics approach, we described the emergence and dispersion of the lineage ST758, and also studied the antibiotic resistance mechanisms present in strains from this ST. This lineage emerged rather recently (about 22 years ago) and since then it has spread in many countries in Latin and North America; notably, strains from this lineage are not only carbapenem resistant but also multidrug resistant. In terms of acquired oxacillinases (OXAs), this lineage presents the worldwide-distributed OXA-23-like family but also the less disseminated OXA-40-like family [24].

Outbreaks caused by MDR *A. baumannii* have been identified in several ICUs worldwide, and the majority only susceptible to colistin [23]. Risk factors for MDR *A. baumannii* infections have been identified: previous hospitalization (the longer the stay, the higher the risk), use of invasive devices, surgical procedures, immunosuppression, admission to the ICU, mechanical ventilation, previous use of antibiotics, and colonization by this bacterium [3, 25]. In our series, 46% of patients had been hospitalized during the previous three months, 28% had used broad-spectrum antimicrobials, 83% had been admitted to the ICU, with a mean length of 16 days in the ICU and a mean length of 18 days in the hospital prior to *A. baumannii* isolation.

During the outbreak, prevention measures were implemented and others were reinforced; the measures included: a single nurse per patient, a single cleaning material quit per each

room, exhaustive daily cleaning of all ventilation equipment, hand hygiene program reinforcement, strict contact isolation, chlorhexidine oral washing three times per day, and patient daily bathing with chlorhexidine. Antimicrobial control policies were implemented in year 2013. Also, for patients transferred from other hospital initial cultures from nares, axillary, perineal region and draining sites were taken. All the measures contributed to end the outbreak in year 2014. Environmental and tap water surveillance was also performed during the outbreak period (44 samples taken in 2011, and 111 in 2012). Only one tap water sample (isolated in Feb 2012) was positive for MDR *A. baumanniii* ST758 profile V at the hematopoietic stem cell transplant unit. No patient in this unit was diagnosed with MDR *A. baumanniii* infection.

Multiple studies have reported poor outcomes of MDR *A. baumannii* infection with high mortality rates and long hospital stays directly attributed to infection [26]. A study conducted in China found a crude and 30-day in-hospital mortality rate of MDR *A. baumannii* bacteremia of 21.2% and 12.7% respectively, lower than other reports in which high mortality rates ranged from 29% to 63.5%. This difference needs to be clarified as the severity of diseases and virulence factors are closely related to mortality [27]. A previous study carried out in Mexico reported 14.5% mortality [28]; this study included susceptible and resistant *A. baumannii* strains in a tertiary care teaching hospital; 22% of patients were not hospitalized in the ICU, which reflects less ill patients compared with this series. Mortality reported in this study was 20% in the first 72 hours, and 50% in the first month, in which patients with bacteremia had the highest mortality (89%). These results were similar to a retrospective cohort from Israel that reported increased mortality in patients with bacteremia and concomitant pneumonia [29]. As expected, the difference in mortality was significant when we compared colonized patients (20%) with infected patients (68.2%). Renal failure and SOFA score >7 have been documented as independent risk factors for mortality in the ICU [27, 30]. We did not find differences in 30-day mortality when SOFA scores were analyzed at ICU admission, but we found differences when this score was calculated the day when culture was obtained (SOFA ≥10 was calculated in 30.2% of patients who died in the first month, compared with 9.4% who were alive in the same period of time, $p = 0.02$).

Although we did not estimate costs, it is very important to consider the increase in costs for the care of MDR organisms outbreaks. In France, the costs observed in a 17-week outbreak was close to 500,000 US dollars [4].

The study has some limitations: we included patients from a single tertiary-care referral cancer center, that does not allow to extrapolate the data to other scenarios. We also were not able to get information from resistance isolates of the hospital where the index patient was hospitalized previous to the transfer to our institution; we were able to establish through molecular epidemiology the original source of the strain.

However, the strengths are that it allows establishing some risk factors associated with infection and mortality from this pathogen, and we were able to demonstrate the clonal relation in most all the strains of this outbreak.

## Conclusions

The molecular epidemiology of this outbreak highlights the threat that represents the transfer of colonized patients with MDR strains from another institutions, and emphasizes the importance to adhere to strict preventive measures, particularly hand hygiene programs, and contact isolation in patients with MDR strains [30]. Although some of the strains collected had different plasmid patterns and did not belong to the main outbreak strain, they were obtained from patients that had been treated in other hospitals, which shows the constant menace of introducing MDR strains into a hospital.

## Supporting information

**S1 File. A method to obtain plasmid profiles from Acinetobacter.**
(DOCX)

**S1 Data.**
(PDF)

## Acknowledgments

Microbiology laboratory staff.

## Author Contributions

**Conceptualization:** Patricia Cornejo-Juárez, Patricia Volkow-Fernández.

**Funding acquisition:** Santiago Castillo-Ramírez.

**Investigation:** Patricia Cornejo-Juárez, Consuelo Velázquez-Acosta, David Martínez-Oliva, Patricia Volkow-Fernández.

**Methodology:** Miguel Angel Cevallos, Semiramis Castro-Jaimes, Angeles Pérez-Oseguera, Frida Rivera-Buendía.

**Resources:** Miguel Angel Cevallos.

**Supervision:** Santiago Castillo-Ramírez, Patricia Volkow-Fernández.

**Validation:** Miguel Angel Cevallos, Santiago Castillo-Ramírez, Angeles Pérez-Oseguera.

**Writing – original draft:** Patricia Cornejo-Juárez.

**Writing – review & editing:** Patricia Volkow-Fernández.

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
