## [Decision Letter · Decision Letter 0]

11 Feb 2020

PONE-D-19-33916

High mortality in an outbreak of multidrug resistant Acinetobacter baumannii introduced to an oncological hospital by a patient transferred from a general hospital.

PLOS ONE

Dear Dra. Volkow-Fernández,

Thank you for submitting your manuscript to PLOS ONE. After careful consideration, we feel that it has merit but does not fully meet PLOS ONE’s publication criteria as it currently stands. Therefore, we invite you to submit a revised version of the manuscript that addresses the points raised during the review process.

We would appreciate receiving your revised manuscript by Mar 27 2020 11:59PM. To enhance the reproducibility of your results, we recommend that if applicable you deposit your laboratory protocols in protocols.io, where a protocol can be assigned its own identifier (DOI) such that it can be cited independently in the future. For instructions see: http://journals.plos.org/plosone/s/submission-guidelines#loc-laboratory-protocols

We look forward to receiving your revised manuscript.

Kind regards,

Monica Cartelle Gestal, PhD

Academic Editor

PLOS ONE

Journal Requirements:

2. In your ethics statement in the manuscript and in the online submission form, please provide additional information about the patient records used in your retrospective study. Specifically, please ensure that you have discussed whether all data were fully anonymized before you accessed them and/or whether the IRB or ethics committee waived the requirement for informed consent.

Reviewers' comments:

Reviewer's Responses to Questions

**Comments to the Author**

1. Is the manuscript technically sound, and do the data support the conclusions?

Reviewer #1: Partly

Reviewer #2: Partly

2. Has the statistical analysis been performed appropriately and rigorously? 

Reviewer #1: Yes

Reviewer #2: Yes

3. Have the authors made all data underlying the findings in their manuscript fully available?

Reviewer #1: Yes

Reviewer #2: Yes

4. Is the manuscript presented in an intelligible fashion and written in standard English?

Reviewer #1: No

Reviewer #2: Yes

5. Review Comments to the Author

Reviewer #1: The manuscript entitled “High mortality in an outbreak of multidrug resistant Acinetobacter baumannii introduced to an oncological hospital by a patient transferred from a general hospital.” by Volkow-Fernández et al. reports an outbreak of multidrug resistant (MDR) A. baumannii

In my opinion the manuscript is of interest from an epidemiological and clinical point of view but the introduction, methods and discussion section are limited, I recommend presenting this manuscript in a better fashion. There are many errors in the document presented.

“We analyzed all MDR A. baumannii strains” How many?

“During the study period, 108 patients with MDR A. baumannii isolates were identified” 1 strain per patient?

“Fig 1a and 1b.” No figures in the manuscript

“Of all the isolates obtained during the outbreak” How many?

The antibiotic resistance profile of the strains?

Did you know what kind of mechanism of resistance was behind the strains?

Please provide a comment in the discussion about detection of these strains in the micro lab and infection control.

What interventions were made in response to the outbreak

In colistin and Tigecycline treatment please include the dose and time according site of infection

Please include in the results and discussion the different ST found in the study

Please rewrite limitations of the study

Reviewer #2: In this manuscript, the authors describe an outbreak of multidrug resistant Acinetobacter baumannii infection in a tertiary care cancer hospital in Mexico City. They describe the clinical features, outcomes, and molecular epidemiology of this outbreak.

Major comments

The clinical part of this work is complete. However, the absence of data concerning the antibiotic resistance profile for each strain is regrettable and must be shown since the antimicrobial susceptibility was determined, as described in the Methods part.

Furthermore, also regrettable, there is no figure with the experimental results of plasmid profile (agarose gels). It will be of interest to visualize these different profiles.

Specific comments

It is stated in the abstract that “we documented an outbreak of an MDR A. baumannii strain identified initially in 2011 ... “ : Is there any clear evidence that all the strains isolated until 2015 are related to the index case ?

It could be of interest to show the relation between clinical data (localization, classification (infected or colonized), malignancies) and the sequence typing of the strain.

Why do the authors performed univariate and multivariate analysis ?

At the end of the Results part, it is stated that “98 had the same sequence type ST758”. However, in the S1 table, it appears that 99 strains exhibit this ST. Could the authors clarify this point ?

The quality of the figures is poor. Furthermore, a detailed legend must be added for each figure.

6. PLOS authors have the option to publish the peer review history of their article (what does this mean?). If published, this will include your full peer review and any attached files.

Reviewer #1: Yes: Eduardo VIllacìs

Reviewer #2: No

---

## [Author Response · Author response to Decision Letter 0]

14 Mar 2020

Reviewer 1. The manuscript entitled “High mortality in an outbreak of multidrug resistant Acinetobacter baumannii infection introduced to an oncological hospital by a patient transferred from a general hospital.” By Volkow-Fernández et al. reports an outbreak of multidrug resistant (MDR A. baumannii).

In my opinion the manuscript is of interest from an epidemiological and clinical point of view but the introduction, methods and discussion section are limited, I recommend presenting this manuscript in a better fashion. There are many errors in the documented presented. 

1. “We analyzed all MDR A. baumannii strains”. How many?

We included in methods the following (page 4, 2nd paragraph) 

In March 2011, we isolated the first strain of MDR A. baumannii; since then, multiple isolates from different sources of hospitalized patients at the ICU were recovered. No MDR A. baumanii strains had been isolated previously in the hospital. During 2011 to 2012, 78 isolates were identified (73.6%), decreasing the number of isolates recovered in the coming three years, until 2015. 

2. During the study period, 108 patients with MDR A. baumannii isolates were identified 1 strain per patient?

We completed the phrase to make clear that was one strain per patient included (Page 4 line 11-12): 

From January 2011 to December 2015, 106 patients with MDR A. baumannii isolates were identified, all were included in the study.

3. Fig. 1a and 1b. No figures in the manuscript.

We checked and all figures were included in the built pdf form at submission.

4. “Of all the isolates obtained during the outbreak” How many?

This information was incorporated in the methods section (page 4, line 11) also commented in in the answer of question 2

5. The antibiotic resistance profile of the strains? 

This information was incorporated in the text (Page 6, first paragraph). 

All the strains included in this study, were resistant to all previous antibiotics families except tetracyclines and colistin.

6. Did you know what kind of mechanism of resistance was behind the strains?

No, we have not investigated it. 

7. Please provide a comment in the discussion about detection of these strains in the micro lab and infection control. What interventions were made in response to the outbreak.

A comment was included about the detection in the microbiology laboratory and the measures taken to control the outbreak.

We incorporated in page 18, 2nd paragraph.:

During the outbreak, prevention measures were implemented and others were reinforced; the measures included: a single nurse per patient, a single cleaning material quit per each room, exhaustive daily cleaning of all ventilation equipment, hand hygiene program reinforcement, strict contact isolation, clorhexidine oral washing three times per day, and patient daily bathing with clorhexidine. Antimicrobial control policies were implemented in year 2013. Also, for patients transferred from other hospital initial cultures from nares, axillary, perineal region and draining sites were taken. All the measures contributed to end the outbreak in year 2014. Environmental and tap water surveillance was also performed during the outbreak period (44 samples taken in 2011, and 111 in 2012). Only one tap water sample (isolated in Feb 2012) was positive for MDR A. baumanniii ST758 profile V at the hematopoietic stem cell transplant unit. No patient in this unit was diagnosed with MDR A. baumanniii infection.

8. In colistin and tigecycline treatment please include the dose and time according site of infection.

The information was included (Page 13, 2nd paragraph).

The mean days for colistin, the mean days were 12 ± 6 days (no differences between type of infection). Two patients received tigecycline (one as monotherapy and one combined with meropenem), for 7 and 14 days each one.

6. Please include in the results and discussion the different MLST found in the study.

We have incorporated MLST findings in the results section (page 8, 2nd paragraph):

Of all the isolates obtained during the outbreak, 99 had the same sequence type MLST 758; 94 had the same plasmid profile as the one recovered from the index case, two had the same plasmid an extra plasmid, and two others were also MLST 758 but had different plasmid profiles. Seven had different MLST (four of them proceeded from infected patients and had been hospitalized in another hospital within the previous 60 days). 

In the discussion section (Page 16, 2nd paragraph:

In this series, we included 99 patients with an MLST 758 strain: 95 with the same plasmid profile (V), two with plasmid XVI- very similar to V, and two with another plasmid profile. The remaining seven isolates belonged to different MLSTs with different plasmids patterns; all of them have been hospitalized recently. They were probably colonized with the MDR strain when they arrived, and it became invasive during hospitalization due to therapeutic procedures. 

This MLST has been identified previously in Mexican hospitals and in other countries of Latin America. The MLST758 is an understudied lineage that does not belong to the international clones I and II; it has spread to Canada, Mexico, Honduras, and Colombia. This lineage represents a source of genetic diversity of MDR and XDR isolates that have not been studied [12, 13].

7. Please rewrite limitations of the study

We incorporate a new version of the limitations of the study. (Page 19, last paragraph).

The study has some limitations: we included patients from a single tertiary-care referral cancer center, that does not allow to extrapolate the data to other scenarios. We also were not able to get information from resistance isolates of the hospital where the index patient was hospitalized previous to the transfer to our institution; we were able to establish through molecular epidemiology the original source of the strain.

However, the strengths are that it allows establishing some risk factors associated with infection and mortality from this pathogen, and we were able to demonstrate the clonal relation in most all the strains of this outbreak. 

Reviewer #2: In this manuscript, the authors describe an outbreak of multidrug resistant Acinetobacter baumannii infection in a tertiary care cancer hospital in Mexico City. They describe the clinical features, outcomes, and molecular epidemiology of this outbreak.

Major comments

The clinical part of this work is complete. However, the absence of data concerning the antibiotic resistance profile for each strain is regrettable and must be shown since the antimicrobial susceptibility was determined, as described in the Methods part.

1. The information with the susceptibility was reviewed and some information was incorporated (Page 5, last paragraph).

Antibiotic disks included ceftriaxone (30 µg), ceftazidime (30 µg), ciprofloxacin (5 µg), imipenem (10 µg), meropenem (10 µg), gentamicin (10 µg), and sulfamethoxazole/trimethoprim (1.25/23.75 µg) (CLSI). Colistin (10 µg) disks were tested although no parameters had been determined by CLSI. All isolates had an inhibition zone diameter ≥21 mm. [7]. Strains are considered MDR if they showed resistance to more than one agent in ≥3 antimicrobial categories such as aminoglycosides, antipseudomonal carbapenems and/or cephalosporins, fluoroquinolones, beta-lactam-beta-lactamase inhibitors, folate pathway inhibitors, and tetracyclines (Magiorakos criteria) [6]. All the strains included in this study, were resistant to all previous antibiotics families except tetracyclines and colistin.

2. Furthermore, also regrettable, there is no figure with the experimental results of plasmid profile (agarose gels). It will be of interest to visualize these different profiles.

The picture of an agarose gel showing the different profiles was included as figure 2.

Specific comments

3. It is stated in the abstract that “we documented an outbreak of an MDR A. baumannii strain identified initially in 2011 ... “ : Is there any clear evidence that all the strains isolated until 2015 are related to the index case ?

The same strain with the same plasmid profile isolated from the index case, that was a patients that had been transferred from a General Hospital to our Institution, was found in 100% of the isolates from the years 2011 to 2012, and in 93.4% from the whole strains. 

4. It could be of interest to show the relation between clinical data (localization, classification (infected or colonized), malignancies) and the sequence typing of the strain.

Ninety-nine of 106 strains had the same MLST, so only 7 patients had a different strain, we considered that the group is too small to do an analysis. 

5. Why do the authors performed univariate and multivariate analysis ?

The uni- and multivariate analyses were performed to assess clinical risk factors related to mortality and for colonization vs. infection.

6. At the end of the Results part, it is stated that “98 had the same sequence type ST758”. However, in the S1 table, it appears that 99 strains exhibit this ST. Could the authors clarify this point ?

There were 98 strains plus the index case, as the way it was written created confusion, we have modified the manuscript.

In the abstract section: 

The index case, identified by molecular epidemiology, was a patient with a drain transferred from a hospital outside Mexico City. Ninety-eight additional cases had the same (MLST 758, of which 94 also had the same plasmid profile, two had an extra plasmid, and two had a different plasmid. The remaining seven isolates belonged to different MLSTs. 

In the result section (Page 8, 2nd paragraph):

Of all the isolates obtained during the outbreak, 99 had the same sequence type MLST 758; 94 had the same plasmid profile as the one t recovered from the index case, two had the same plasmid an extra plasmid, and two others were also MLST 758 but had different plasmid profiles. Seven had different MLST (four of them proceeded from infected patients and had been hospitalized in another hospital within the previous 60 days).

7. The quality of the figures is poor. Furthermore, a detailed legend must be added for each figure.

A new version of the figure much clearer with the corresponding figure legend (page 8, 3rd paragraph).

---

## [Decision Letter · Decision Letter 1]

2 Apr 2020

PONE-D-19-33916R1

High mortality in an outbreak of multidrug resistant Acinetobacter baumannii introduced to an oncological hospital by a patient transferred from a general hospital.

PLOS ONE

Dear Dra. Volkow-Fernández,

Thank you for submitting your manuscript to PLOS ONE. After careful consideration, we feel that it has merit but does not fully meet PLOS ONE’s publication criteria as it currently stands. Therefore, we invite you to submit a revised version of the manuscript that addresses the points raised during the review process.

Your revisions have significantly improved this manuscript, however, they are asking for more. We would appreciate receiving your revised manuscript by May 17 2020 11:59PM. To enhance the reproducibility of your results, we recommend that if applicable you deposit your laboratory protocols in protocols.io, where a protocol can be assigned its own identifier (DOI) such that it can be cited independently in the future. For instructions see: http://journals.plos.org/plosone/s/submission-guidelines#loc-laboratory-protocols

We look forward to receiving your revised manuscript.

Kind regards,

Monica Cartelle Gestal, PhD

Academic Editor

PLOS ONE

Reviewers' comments:

Reviewer's Responses to Questions

**Comments to the Author**

1. If the authors have adequately addressed your comments raised in a previous round of review and you feel that this manuscript is now acceptable for publication, you may indicate that here to bypass the “Comments to the Author” section, enter your conflict of interest statement in the “Confidential to Editor” section, and submit your "Accept" recommendation.

Reviewer #1: (No Response)

Reviewer #2: All comments have been addressed

2. Is the manuscript technically sound, and do the data support the conclusions?

Reviewer #1: Partly

Reviewer #2: Yes

3. Has the statistical analysis been performed appropriately and rigorously? 

Reviewer #1: Yes

Reviewer #2: N/A

4. Have the authors made all data underlying the findings in their manuscript fully available?

Reviewer #1: No

Reviewer #2: Yes

5. Is the manuscript presented in an intelligible fashion and written in standard English?

Reviewer #1: No

Reviewer #2: Yes

6. Review Comments to the Author

Reviewer #1: In my opinion, the authors solved many of the questions and the manuscript is of interest from a clinical and epidemiological point of view. However, it is important to improve the manuscript in scientific content and the presentation to cover the standards required for publication.

Please, I recommend to include line numbers

[5]. “This pathogen is able to grow at various temperatures and pH conditions, so it has the ability to persist in either moist or dry conditions in the hospital environment, thereby contributing to transmission” [5] Please include more references in the study and avoid same citation in a paragraph

“identification and antimicrobial susceptibility were determined using the Gram-Negative Identification Panel (BDPhoenixTM automated system, New Jersey, USA).” Please include the MIC values.

“DNA was extracted from a representative strain of each patter” How many AMR patterns did you find?

“A typical pulsed-field gel electrophoresis analysis of selected isolates of A. baumannii and representative plasmid profiles is shown in figure 2.” Please include PFGE in methods section.

“First, rpoB’s Zone-1[9] was amplified to confirm taxonomic status” Amplified and sequenced?

This MLST has been identified previously in Mexican hospitals and in other countries of Latin America. Please include more references to improve the discussion.

It would be important to characterize the resistance mechanisms found in ST758, and if possible, to sequence the strain and the plasmids that will show relevant information to complement the clinical part of the study.

Reviewer #2: (No Response)

7. PLOS authors have the option to publish the peer review history of their article (what does this mean?). If published, this will include your full peer review and any attached files.

Reviewer #1: No

Reviewer #2: No

---

## [Author Response · Author response to Decision Letter 1]

30 Apr 2020

6. Review Comments to the Author

Reviewer #1: In my opinion, the authors solved many of the questions and the manuscript is of interest from a clinical and epidemiological point of view. However, it is important to improve the manuscript in scientific content and the presentation to cover the standards required for publication.

1. Please, I recommend to include line numbers

We included line numbers. 

2. [5]. “This pathogen is able to grow at various temperatures and pH conditions, so it has the ability to persist in either moist or dry conditions in the hospital environment, thereby contributing to transmission” [5] Please include more references in the study and avoid same citation in a paragraph.

Answer: We have included 3 references related to the capacity of the pathogen to survive and multiple in diverse conditions. Page 3. Line 67.

- Baumann P. Isolation of Acinetobacter from soil and water. J Bacteriol. 1968;96(1):39–42

- Warskow AL, Juni E. Nutritional requirements of Acinetobacter strains isolated from soil, water, and sewage. J Bacteriol. 1972;112(2):1014–1016.

- Jawad A, Heritage J, Snelling AM, Gascoyne-Binzi DM, Hawkey PM. Influence of relative humidity and suspending menstrua on survival of Acinetobacter spp. on dry surfaces. J Clin Microbiol. 1996;34(12):2881–2887.

3. “identification and antimicrobial susceptibility were determined using the Gram-Negative Identification Panel (BDPhoenixTM automated system, New Jersey, USA).” Please include the MIC values

MICs values were included. Page 6, lines 126-128.

4. “DNA was extracted from a representative strain of each patter” How many AMR patterns did you find?

There was the same antimicrobial resistance pattern in all MLST758 samples (n=99, 93.4%). In the remaining 7 strains, there was a minimal difference in the MICs of different antibiotics, including quinolones, beta-lactams, and carbapenems, however they were considered resistant according to CLSI.

5. “A typical pulsed-field gel electrophoresis analysis of selected isolates of A. baumannii and representative plasmid profiles is shown in figure 2.” Please include PFGE in methods section.

There was a mistake in this statement, because we did not perform PFGE. The sentence was changed by: “Representative plasma profiles of selected Acinetobacter strains are shown in figure 2.” Page 8, line 179.

6. “First, rpoB’s Zone-1[9] was amplified to confirm taxonomic status” Amplified and sequenced?

Yes, amplified and sequenced. Sequenced was added in the sentence. Page 6, line 132. A method to obtain plasmid profiles from Acinetobacter was included as Supplement.

7. This MLST has been identified previously in Mexican hospitals and in other countries of Latin America. Please include more references to improve the discussion.

We included three references in this part. Page 16, line 270.

- Alcántar-Curiel MD, Rosales-Reyes R, Jarillo-Quijada MD, et al. Carbapenem-Resistant Acinetobacter baumannii in Three Tertiary Care Hospitals in Mexico: Virulence Profiles, Innate Immune Response and Clonal Dissemination. Front Microbiol 2019; 10: 2116. Published 2019 Sep 20. doi:10.3389/fmicb.2019.02116

- Tamayo-Legorreta EM, Garza-Ramos U, Barrios-Camacho H, et al. Identification of OXA-23 carbapenemases: novel variant OXA-239 in Acinetobacter baumannii ST758 clinical isolates in Mexico. New Microbes New Infect 2014; 2: 173–4. doi:10.1002/nmi2.60

- Vanegas JM, Higuita LF, Vargas CA, Cienfuegos AV, Rodríguez EA, Roncancio GE, Jiménez JN. [Carbapenem-resistant Acinetobacter baumannii causing osteomyelitis and infections of skin and soft tissues in hospitals of Medellín, Colombia]. Biomedica 2015; 35:522–30.

8. It would be important to characterize the resistance mechanisms found in ST758, and if possible, to sequence the strain and the plasmids that will show relevant information to complement the clinical part of the study.

Thanks for the comment. We have very recently published a manuscript (see reference below) where we not only characterised the resistance mechanisms in ST758 but also studied the phylogeography of this lineage. Of note, in this study more than 20 ST758 genomes were considered. To enrich our manuscript, we have included a couple of sentences in the discussion (page 18, first paragraph) talking about the resistance mechanisms found in ST758.

Recently, employing a population genomics approach, we described the emergence and dispersion of the lineage ST758, and also studied the antibiotic resistance mechanisms present in strains from this ST (REF mSphere article). This lineage emerged rather recently (about 22 years ago) and since then it has spread in many countries in Latin and North America; notably, strains from this lineage are not only carbapenem resistant but also multidrug resistant. In terms of acquired oxacillinases (OXAs), this lineage presents the worldwide-distributed OXA-23-like family but also the less disseminated OXA-40-like family. Page 18, lines 298-304.

The reference was included: Graña-Miraglia L, Evans BA, López-Jácome LE, et al. Origin of OXA-23 Variant OXA-239 from a Recently Emerged Lineage of Acinetobacter baumannii International Clone V. mSphere. 2020;5(1):e00801-19. Published 2020 Jan 8. doi:10.1128/mSphere.00801-19

Reviewer #2: (No Response)

---

## [Decision Letter · Decision Letter 2]

2 Jun 2020

High mortality in an outbreak of multidrug resistant Acinetobacter baumannii introduced to an oncological hospital by a patient transferred from a general hospital.

PONE-D-19-33916R2

Dear Dr. Volkow-Fernandez,

We are pleased to inform you that your manuscript has been judged scientifically suitable for publication and will be formally accepted for publication once it complies with all outstanding technical requirements.

With kind regards,

Monica Cartelle Gestal, PhD

Academic Editor

PLOS ONE

Additional Editor Comments (optional):

Reviewers' comments:

Reviewer's Responses to Questions

**Comments to the Author**

1. If the authors have adequately addressed your comments raised in a previous round of review and you feel that this manuscript is now acceptable for publication, you may indicate that here to bypass the “Comments to the Author” section, enter your conflict of interest statement in the “Confidential to Editor” section, and submit your "Accept" recommendation.

Reviewer #1: All comments have been addressed

Reviewer #2: All comments have been addressed

2. Is the manuscript technically sound, and do the data support the conclusions?

Reviewer #1: Yes

Reviewer #2: Yes

3. Has the statistical analysis been performed appropriately and rigorously? 

Reviewer #1: Yes

Reviewer #2: N/A

4. Have the authors made all data underlying the findings in their manuscript fully available?

Reviewer #1: Yes

Reviewer #2: Yes

5. Is the manuscript presented in an intelligible fashion and written in standard English?

Reviewer #1: Yes

Reviewer #2: Yes

6. Review Comments to the Author

Reviewer #1: Authors solved the questions and the manuscript is of interest Please, I recommend minimal changes

Line 179. Please, italics in Acinetobacter and correct “plasma” plasmid

Reviewer #2: The authors have adressed my questions.

Did the authors try to characterize the antimicrobial resistance mechanisms in the index case ? It would be a very interesting data. Are the resistance determinants plasmid-borne ?

P. 8, line 179 : It is "plasmid" profile, not "plasma" profile

7. PLOS authors have the option to publish the peer review history of their article (what does this mean?). If published, this will include your full peer review and any attached files.

Reviewer #1: No

Reviewer #2: No

---

## [Editor Report · Acceptance letter]

12 Jun 2020

PONE-D-19-33916R2 

High mortality in an outbreak of multidrug resistant *Acinetobacter baumannii* infection introduced to an oncological hospital by a patient transferred from a general hospital. 

Dear Dr. Volkow-Fernández:

I'm pleased to inform you that your manuscript has been deemed suitable for publication in PLOS ONE. Congratulations! Your manuscript is now with our production department. 

Kind regards, 

on behalf of

Dr. Monica Cartelle Gestal 

Academic Editor

PLOS ONE